# Pre-Implementation Evaluation of a Community-Based Surveillance System for Migrants' Sexual Health in Chile

Constanza Adrian Parra [1], Valeria Stuardo Ávila [2,3,*], Kenny Low Andrade [4], Cristian Lisboa Donoso [5], Débora Solís [6], Danilo Gómez [7], Evelyn Cortés [8], Cecilia Bustos Ibarra [9], Paola Contreras Hernández [10], Jaime Barrientos Delgado [11] and Mercedes Carrasco-Portiño [12]

1  Doctoral Program in Biomedical Research Methodology and Public Health, Department of Pediatrics, Obstetrics, and Gynecology and Preventive Medicine and Public Health, Universitat Autònoma de Barcelona, 08193 Barcelona, Spain; constanzaadrian4340@gmail.com
2  Institute of Public Health, Universidad Andrés Bello, Santiago 7550000, Chile
3  School of Public Health, Faculty of Medicine, University of Chile, Santiago 8380000, Chile
4  Independent Researcher, Santiago 8320000, Chile; klowandrade@gmail.com
5  Department of Health Sciences, School of Dentistry, Universidad Autonoma de Chile, Santiago 7500000, Chile; cristian.lisboa@ug.uchile.cl
6  APROFA, Santiago 7500000, Chile; dsolis@aprofa.cl
7  Chilean Red Cross Local Branch in Antofagasta, Antofagasta 1260000, Chile; filial.antofagasta@cruzroja.cl
8  Crealuz Centro Cultural y de Desarrollo Personal, Antofagasta 1240000, Chile; mayacian.b.a@gmail.com
9  Department of Social Work, School of Social Sciences, Universidad de Concepción, Concepción 4070386, Chile; cecbustos@udec.cl
10  Department of Education, Universidad de Tarapacá, Arica 1000000, Chile; pcontrerash@academicos.uta.cl
11  Department of Psychology, Universidad Alberto Hurtado, Santiago 8320000, Chile; jbarrientos@uahurtado.cl
12  Department of Obstetrics and Childcare, School of Medicine, Universidad de Concepción, Concepción 4070386, Chile; mecarrasco@udec.cl
*  Correspondence: valeria.stuardo@unab.cl

## Abstract

The increasing migration phenomenon and its impact on sexual health highlight the urgency of improving access to preventive services and developing responsive surveillance mechanisms. This study aims to describe the context and define the implementation circuits of a Community-Based Surveillance System (CBSS) focused on social epidemiological aspects related to sexual health in Chile's migrant population. A two-phase qualitative design was employed: Phase 1 involved formative research, and Phase 2 focused on the design of CBSS implementation circuits. The formative phase led to the selection and characterization of three community-based organizations (CBOs)—two in Santiago and one in Antofagasta—and two primary healthcare centers (PHCs). Findings revealed heterogeneity in institutional capacities, limited coordination between CBOs and the health system, and a high level of willingness to participate. PHCs showed comparable profiles. Based on this, differentiated operational circuits were co-designed and adapted with stakeholders, and formalized through site-specific implementation manuals. This pre-implementation evaluation helped identify critical contextual barriers and generate tailored strategies for CBSS deployment. The active involvement of local actors is essential to ensuring the contextual relevance, institutional acceptability, and future sustainability of the proposed model. These insights offer transferable learning for the design of health interventions in underserved and structurally constrained settings.

**Keywords:** sexual health; community-based surveillance; migrants; implementation research

## 1. Introduction

According to the Lancet Commission on Migration and Health [1], there are key challenges to ensuring safe and healthy migration. Migratory trajectories, ranging from the decision to emigrate and transit to processes of regularization or return, are often marked by situations of violence [1,2], which disproportionately affect groups experiencing heightened vulnerability. Women, for example, are often subjected to sexual violence and exploitation, while gender-diverse individuals additionally face stigma and discrimination [1]. These conditions increase exposure to risk factors for sexually transmitted infections (STIs), among others, often accompanied by limited or no access to healthcare services [1,3,4]. Therefore, ensuring access to preventive healthcare services in destination countries, as well as developing indicators that account for specific needs related to sexual health (SH), constitutes a priority for public health [1,5,6].

Migration in Chile has increased significantly over the last decade. In 2023, migrants accounted for 9.6% of the population, representing a 46.8% increase compared to 2018 [7]. The metropolitan area, located in the center of Chile and covering its capital, Santiago de Chile, has the largest proportion of foreigners, with 56.8% of the total estimated for 2023, followed by Antofagasta (6.7%) in the north of the country [7]. Additionally, there has been a sharp rise in the proportion of migrants with irregular migration status, which grew from 0.8% in 2018 to 17.6% in 2023 [5]. More than 70% of the migrant population originates from other Latin American countries, primarily Venezuela (38%) and Peru (13.6%) [7].

In response to this population's health needs and to ensure their access to care, Presidential Decree No. 67 of 2016 mandated that the public health system had to provide medical care to migrants, regardless of their documentation or residence status. This regulation also stipulated their inclusion as beneficiaries of the National Health Fund (FONASA, for its Spanish acronym) and their enrollment in primary, secondary, or tertiary healthcare levels [8]. However, several barriers to accessing and utilizing healthcare services still exist for this population [9], as evidenced by 16.4% of migrants lacking health insurance (compared to 1.7% of the local population) and an equal proportion not receiving care for recent health issues, vs. 9.5% among Chilean-born individuals [10]. In parallel, there has been a re-emergence of the HIV epidemic in Chile, with an incidence of 0.25 cases per 1000 inhabitants in 2022 [11], along with an increase in cases of syphilis and gonorrhea [12,13]. New HIV diagnoses in migrants grew from 20.2% to 33.8% between 2017 and 2021 [14], suggesting an increase in SH needs among migrants that have not been fulfilled by the formal healthcare system.

However, conventional epidemiological surveillance systems, which focus primarily on biomedical indicators, fail to capture the social and cultural factors underlying these disparities, limiting the design of contextualized interventions. Community-based research offers a critical approach to addressing these gaps, as it enables access to populations and issues that are frequently overlooked by traditional methods [15,16]. By actively involving communities, this research generates essential evidence to inform equity-focused policies and programs, particularly in sexual health [17]. Within this framework, community-based surveillance (CBS), understood as an active process of community participation in the detection, reporting, response, and follow-up of health events in the community, involves the continuous and systematic collection of data [18] and can provide sensitive information for the timely detection of diseases or aspects related to health inequalities [19].

In Chile, community-based organizations (CBOs) often serve as the first point of care for vulnerable populations, especially migrants with irregular status. However, the absence of a system for collecting and analyzing socio-epidemiological and cultural information restricts access to key data needed for decision-making and the effective implementation of sexual health policies. Within this framework, implementation research offers an integrative

approach that combines study and practice, fostering collaboration among communities, implementers, researchers, and policymakers to enhance the execution of health policies with greater equity, efficiency, and sustainability [20], where monitoring and evaluation are essential for adapting and iterating these interventions [21].

This work is part of the COSMIC project (FONDECYT, Regulation No. 1220371), which aims to develop a system to monitor social epidemiological and cultural aspects a related to SH and communicable diseases in the migrant population in Chile, as well as to facilitate their linkage with primary healthcare centers (PHCs), within the framework of implementation research [22]. As a foundational step, a set of socio-epidemiological indicators of sexual health was previously developed, understood as social, cultural, economic, and contextual factors that, in interaction with epidemiological patterns, determine the sexual health status of migrant populations [23]. Building on this groundwork, a Community-Based Surveillance System (CBSS) was designed to monitor these indicators in real time. This article evaluates the pre-implementation phase of the CBSS, with the aim of describing its context and defining its implementation circuits, a key component within implementation research.

## 2. Materials and Methods

### 2.1. Design

This pre-implementation evaluation employed a qualitative methodology grounded in implementation research (IR) principles, which emphasize the importance of generating context-specific evidence to inform the adaptation and scaling of interventions in real-world settings. This study was conducted in two sequential phases: (1) formative research, and (2) definition of CBSS implementation circuits.

### 2.2. Study Setting and Context

The study was conducted in Santiago and Antofagasta, cities with the highest proportions of international migrants in Chile [7] and a significant burden of communicable diseases related to SH [14]. Accordingly, the target population consisted of CBOs and PHCs located in these cities, given their strategic relevance for the implementation of a CBSS tailored to the sexual health needs of migrant populations.

### 2.3. Phase 1: Formative Research

Formative research helps define and assess a community of interest to describe relevant attributes for specific public health problems [24]. This phase involved a review of the literature and collection of qualitative data through semi-structured interviews to identify and characterize CBOs and PHCs potentially suitable for CBSS implementation.

#### 2.3.1. Identification of CBOs and PHCs

First, CBOs addressing migration and SH in the selected cities were identified with the literature review (including institutional reports, regulatory frameworks, and legal documents) as well as online and social media searches. The objective was to characterize their activities and establish initial contact.

Subsequently, brief structured interviews were conducted with representatives of these CBOs to present the study objective and apply screening questions aimed at determining whether they addressed health issues—particularly sexual health—and whether they worked directly with migrant populations or included migrants among their beneficiaries.

For PHCs, the selection criteria included being healthcare centers that were part of the Access to Migrant Healthcare Program of the Ministry of Health.

2.3.2. Characterization and Selection of Participants for the COSMIC Project

Once CBOs and PHCs were identified, semi-structured interviews were conducted with their representatives and other key stakeholders (community leaders) in migration and health. The interview guide (Supplementary Material) covered the general characteristics, organizational structure, specific actions related to health and SH, links with other institutions, and prior research involvement. The interviews were recorded, transcribed, and analyzed by deductive thematic analysis. Codes and categories were developed to extract relevant information. For the CBOs and PHCS, the categories of analysis were as follows (Table 1):

**Table 1.** Analytical Categories for the Characterization Process of CBOs and PHCs.

| Institution | Category | Description |
| --- | --- | --- |
| **CBOs** | Administrative characterization | Includes location, facilities, participants, technical capacities, and migrant population served. |
| | Organizational objectives | Identifies general actions and those related to sexual health. |
| | Connection | Explores relationships with other CBOs, public programs, and PHCs. |
| | Community research | Analyzes past research experience, its evaluation, and expectations for future collaboration. |
| **PHCs** | General characteristics | Covers administrative unit, location, healthcare staff, available programs, and services. |
| | Sexual health programs | Describes program origin (MINSAL or local), staff, and healthcare services offered. |
| | Access facilitators | Focuses on enrollment procedures and initial interactions for migrants. |
| | Access barriers | Identifies barriers in accessing sexual health services for migrant population. |
| | Link to CBOs | Examines knowledge of and coordination with CBOs. |
| | Reception of external research | Assesses experiences with and evaluations of external research initiatives. |

Source: Prepared by the authors, COSMIC project.

Following characterization, the most suitable CBOs and PHCs were selected and invited to participate in the COSMIC project, while respecting the ethical considerations in research.

*2.4. Phase 2: Definition of CBSS Implementation Circuits*

The CBSS "Implementation Circuits" are the operational pathways designed to structure user participation at both OBCs and PHCs. These flowcharts serve as tool to define roles, processes, and responsibilities, facilitating a shared understanding and adaptation to local capacities and contextual [23]. This phase involved group interviews with key informants from CBOs and PHCs, followed by a group meeting for analysis, feedback, and consensus on the implementation circuits.

2.4.1. Operational Considerations from Key Informant Interviews

Before implementing the CBSS, six semi-structured group interviews were conducted with key informants from CBOs and PHCs in Santiago and Antofagasta to explore perceived barriers and facilitators. The interview guide (Supplementary Material) addressed topics such as staff training needs in SH for migrants (e.g., cultural relevance and communication), digital platform design, and information flow, among others.

Participants were selected through convenience sampling among COSMIC project partners. The interviews were recorded, transcribed, and analyzed using deductive thematic analysis for qualitative information to identify key implementation considerations.

2.4.2. Design of the Surveillance System Implementation Circuit

The implementation circuits for CBOs and PHCs were developed to support the operationalization of the CBSS. Their design was grounded in conceptual assumptions of CBSS and operational considerations identified through key informant interviews, and was subsequently refined through group review, feedback, and consensus.

## 3. Results

The following results derive from the pre-implementation evaluation of the CBSS, structured in two complementary phases: (1) a formative research phase and (2) the definition of implementation circuits.

### 3.1. Phase 1: Formative Research

3.1.1. Identification of Organizations and PHCs

According to the literature review, based on Chilean legislation (Acts No. 19,418 and No. 20,500), CBOs are private, autonomous, non-profit organizations composed of voluntary members, and governed by formal statutes. As of 2020, they represented 80% of civil society organizations in Chile, with most of them recently established and operating in a decentralized manner. However, only 1% of these were focused on health-related issues [25].

Through online platforms and social media, 35 CBOs working on migration and health were initially identified (22 in Santiago and 13 in Antofagasta). Of these, eight in Santiago and five in Antofagasta participated in the initial characterization process. Interviews with representatives of these organizations explored their institutional work and interest in community-based research. In parallel, 15 key informants (1 from CBOs and 6 from PHCs in Santiago; 3 from CBOs and 5 from PHCs in Antofagasta) were interviewed to gather further insights into their experiences and knowledge.

As for the PHCs, four were identified as particularly relevant due to their high proportion of migrant population: two in Antofagasta and two in Santiago (one in the district of Independencia and one in the district of Santiago Centro).

3.1.2. Characterization and Selection of Participants for the COSMIC Project

A total of 29 interviews were conducted (16 in Santiago and 13 in Antofagasta), which allowed access to representatives of 14 CBOs and 15 key informants. The analysis showed that most CBOs prioritized political activism and the defense of migrants' rights, while health—particularly sexual health—was addressed as a secondary concern, mainly through occasional counseling activities.

Large CBOs, with dedicated headquarters and sufficient resources to pay their members or hire professionals, were a minority within the group studied. Most organizations were smaller, composed of volunteers, and often lacked their own premises. Regarding collaboration, there was substantial interconnection among CBOs focused on defending migrants' rights, along with formal coordination with state institutions—particularly those linked to social development —for the funding of specific projects. However, their engagement with the health system was generally informal and sporadic, primarily to handle emergencies by leveraging personal contacts to facilitate access. Finally, as concerning the reception of external research initiatives, there was little systematic work and a prevailing distrustful attitude toward academic research.

A total of four CBOs were selected, two in Santiago and two in Antofagasta, the latter integrated into a single operational unit to enhance their collective capabilities. The CBOs had their own infrastructure (headquarters), including a room or private space suitable for confidential counseling, conducted activities related to SH, and demonstrated the

technical capacity to deliver such services. Moreover, all expressed a clear willingness to engage in community-based research in collaboration with academic institutions. Table 2 characterizes each of them:

**Table 2.** Characterization of selected community-based organizations.

| Aspects | CBO 1 | CBO 2 | CBO 3 | CBO 4 |
|---|---|---|---|---|
| **Infrastructure** | Headquarters in Santiago, Conchalí district | Headquarters in Santiago, Providencia district | Headquartered in the city of Antofagasta | Based in the city of Antofagasta |
| **Staff** | Volunteers | Employed healthcare personnel | Volunteers | Volunteers |
| **Services available** | Rapid testing for HIV and other STIs and sexual health counseling | Sexual and reproductive healthcare | Emergency assistance and health and wellbeing promotion | Rapid HIV testing and sexual health counseling |
| **Target population** | Gender-diverse individuals | Women | General population | Gender-diverse individuals |
| **Willingness to assist migrants** | Yes | Yes | Yes | Yes |
| **Willingness to work on research projects** | Yes | Yes | Yes | Yes |

Source: Prepared by the authors, COSMIC project.

The characterization of PHCs revealed structural similarities. Family Health Care Centers (CESFAM) operate within defined territories and provide preventive, family centered care. Their smaller counterparts, Community Family Health Care Centers (CECOSF), are administratively dependent on CESFAMs and have fewer staff. Both offer multidisciplinary services. SH care is limited to the National Women's Program, which focuses on pregnancy and family planning. Although counseling is available to support migrants in accessing care and obtaining a Temporary Health Care Number (NIP), significant barriers persist, such as waiting times, insufficient intercultural training, and inadequate social treatment. Engagement with CBOs and research initiatives remains scarce, mainly due to the staff workload.

Two PHCs were selected, one CESFAM in Santiago (Independencia district, PHC_S) and one CECOSF in Antofagasta (PHC_A). Given the structural similarities among centers, selection was primarily based on their expressed willingness to participate in the COSMIC project and their readiness to implement adjustments to improve migrant healthcare services.

Finally, to establish the linkage between each institution and the COSMIC project team, a representative was designated for each CBO and PHC, most of whom were professionals from the health or social sciences fields. These representatives played a key role in coordinating with the project and within their respective institutions.

In conclusion, this phase enabled the characterization of the organizational and healthcare context relevant to CBSS implementation, identifying key limitations such as insufficient infrastructure, limited human resources, and weak linkages between CBOs and the health system. Based on these findings, the most suitable OBCs and PHCs were selected for implementation. This contextual understanding will inform the next phase of the study, allowing for the adaptation of implementation strategies to ensure their relevance and alignment with local capacities.

### 3.2. Phase 2: Definition of Implementation Circuits

Building on the findings of the formative phase, the second stage of this pre-implementation evaluation focused on designing and tailoring the implementation circuits of the CBSS.

### 3.2.1. Operational Considerations from Key Informant Interviews

The information from the CBO and PHC interviews was systematized to identify key aspects for the development of the CBSS implementation circuits. This analysis led to the following results:

- CBOs:

CBOs have similarities and differences in their readiness and installed capacity for the implementation of the CBSS proposed by the COSMIC project (Table 3).

**Table 3.** Results of interviews with CBO members for the development of CBSS implementation circuits.

| Dimension | Description of Findings |
|---|---|
| **Monitoring system functioning** | Most of the CBOs required computer equipment. To enter data in the COSMIC's digital platform, differentiated access profiles had to be set, considering that the person making the initial registration may differ from the one providing care. |
| **Visualization of data generated by COSMIC** | All the CBOs considered the information to be useful, although the perception of its purpose varied, from informing superiors to community outreach or project applications. |
| **Recruitment of participants** | Uncertainty about the inclusion and eligibility criteria was identified, particularly in relation to non-Spanish-speaking migrants and children under 18 years of age. The CBOs also presented different levels of adequacy in terms of physical space and opening hours available for the implementation of CBSS: in general, opening hours were restricted to spontaneous consultations, except for CBO 2, whose staff worked extended hours. |
| **Other common needs of CBOs** | Material for project dissemination, access to the health system and on SH topics were needed, as well as a maximum time of 30 min to monitor indicators and care, and the adequacy of the mechanisms for referral to PHCs by means of a paper form, highlighting the importance of proper coordination with the COSMIC team and PHCs. |

Source: Prepared by the authors, COSMIC project.

- PHCs:

In both PHCs, similarities were found in terms of operation and preferences for implementing the CBSS (Table 4).

**Table 4.** Results of interviews with members of PHCs for the development of CBSS implementation circuits.

| Dimension | Description of Findings |
|---|---|
| **Referral from CBO to PHC** | Referral was made through a paper form with COSMIC identification and the generation of an anonymized code indicating the referral professional. Moreover, PHC_S considered it necessary to incorporate information on the reason for the referral. |
| **Reception of referral** | In both establishments, the reception was made by staff of the Medical-Statistical Guidance Service (SOME, for its Spanish acronym), where user registration at PHC is verified and the NIP for migrants is processed. At PHC_S, the possibility of exclusive care for residents/workers under the jurisdiction of the North Metropolitan Health Service of Santiago [26], their territory of health responsibility, was suggested. |
| **Processing of a temporary health care number** | No PHC reported receiving migrants without an ID card/passport. However, while individuals from the PHC_A were informed of the existence of a legal way to register migrants (Circular N°5) [27], this was not the case for PHC_S. |

**Table 4.** *Cont.*

| Dimension | Description of Findings |
|---|---|
| **Sexual healthcare** | Individuals referred by COSMIC could not be immediately treated at none of the PHCs. However, providing care according to availability was regarded as feasible by PHC_A, whereas PHC_S considered it under its usual schedule. |
| **Data visualization** | Although the healthcare team at both centers did not consider using the data generated by COSMIC, its value to community workers and the migrant health program was recognized [27]. |
| **Coordination with CBO and the COSMIC team** | Both centers showed availability to hold coordination meetings with CBOs and the COSMIC team, preferably coordinated via email or WhatsApp. |
| **Other common needs** | Both PHCs had several operational mechanisms in common, but differed in certain aspects regarding migrant enrollment, as well as in the flexibility or possibility to provide prioritized SH care. |

Source: Prepared by the authors, COSMIC project.

3.2.2. Design of the Implementation Circuit of the Surveillance System

Initial Proposal

The preliminary circuits were informed by conceptual assumptions drawn from the international literature on CBS and public health surveillance systems, as well as national guidelines for primary healthcare in Chile. These initial models proposed a dual-circuit structure—one for CBOs and one for PHCs—reflecting their distinct but complementary roles in surveillance: CBOs as frontline actors in community engagement, early case identification, and linkage to care; and PHCs as institutional nodes for clinical care. The definition of the operational stages also incorporated key considerations derived from the interviews with CBO and PHC staff, which provided strategic insights into infrastructure, referral, information flow, and user eligibility. After analyzing these interviews, relevant components for the CBSS circuit were clearly identified, covering stages from participant recruitment to referral to PHCs. Based on this, a general circuit proposal was developed for CBOs, as shown in Figure 1.

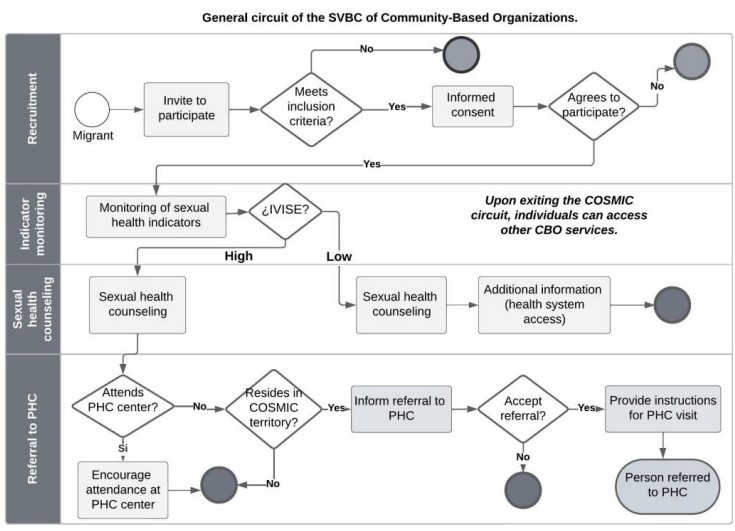

**Figure 1.** General circuit of the CBOs' CBSS.

The proposed circuit was structured into distinct operational stages, each of which is described below:

**Stage 1, recruitment:** takes place from the moment migrants visit each CBO, where compliance with the inclusion criteria is verified. If they meet these requirements, they are invited to participate in the COSMIC project and proceed with signing the informed consent. If the person chooses not to participate, the usual care continues to be provided at each CBO.

**Stage 2, monitoring of social epidemiological indicators of sexual health:** monitoring is carried out through a questionnaire available on the web platform. A community worker generates a unique code for each participant and applies the set of social epidemiological indicators [23]. At the end of the instrument application process, the platform automatically calculates the Individual Vulnerability Index Related to Sexual Health (IVISE, for its Spanish acronym). Regardless of the result obtained, all participants are directed to the next stage of SH counseling.

**Stage 3, sexual health counseling:** SH counseling is offered according to everyone's needs. If the IVISE is low, information on access to the health system is provided, and participation is concluded. If the IVISE is high, referral to a PHC is initiated.

**Stage 4, referral to a PHC:** Migrants are checked to see if they are enrolled in a PHC center. If not, information is provided on how to access one, and the referral process is explained. Individuals may accept or reject the referral. If they accept, they are given the form to facilitate access to the PHC.

A similar process was followed for PHCs, resulting in the development of a general implementation circuit depicted in Figure 2.

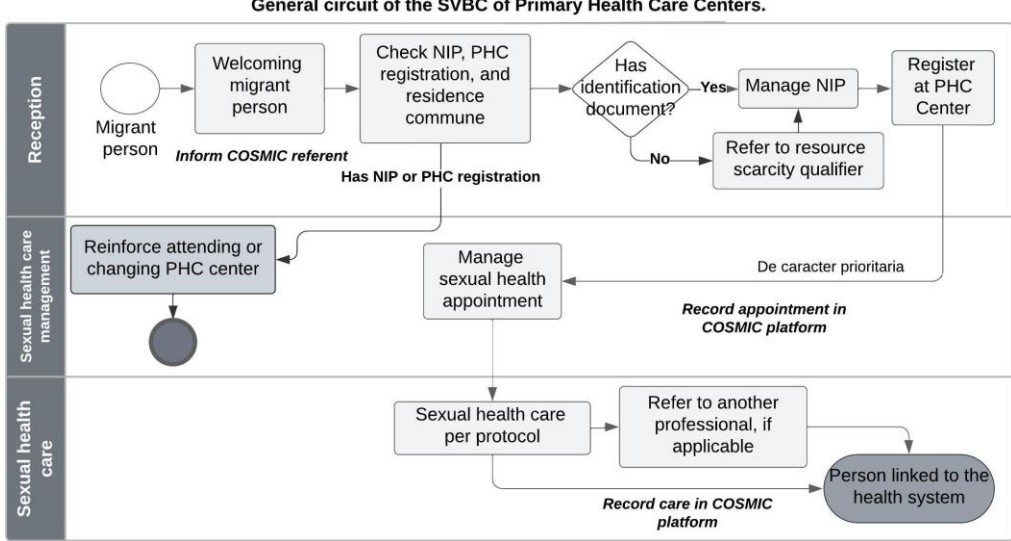

**Figure 2.** General circuit of the PHCs' CBSS.

This circuit includes the following stages:

**Stage 1, reception:** Migrants referred from the CBO visit the PHC with their referral form. The receiving staff verifies their documentation and, if applicable, start the enrollment process in the establishment.

**Stage 2, management of health care appointment:** Once enrolled, an appointment is arranged with a sexual health professional (midwife or matron). It is essential to record these proceedings to ensure traceability of the follow-up process.

**Stage 3, sexual healthcare:** Sexual healthcare is provided and, if required, the patient is referred to another professional for further care, under the principle of comprehensive healthcare.

Participatory Adaptation and Final Implementation Circuits

The initial CBBS implementation circuits were presented to representatives of CBOs and PHCs in a joint working session. This participatory activity aimed to empirically validate the proposed model and generate specific adjustments based on institutional capacities. The session involved a guided review of each stage of the circuits, encouraging participants to share perceived challenges, feasibility constraints, and adaptation suggestions.

The discussions were organized separately for CBOs and PHCs, acknowledging the heterogeneity between these types of institutions. For CBOs, the main challenges identified included staff assignment for participant reception, verification of inclusion criteria, and management of the informed consent process. In PHCs, key issues centered on verifying eligibility for care, standardized user registration on the platform, and scheduling of sexual health appointments. The topics raised during these sessions were directly integrated into the redesign of the implementation circuits, resulting in specific operational adjustments tailored to each institutional context. Table 5 summarizes the main operational challenges and the changes implemented through this participatory process.

**Table 5.** Main operational challenges and changes implemented in the final CBSS implementation circuits.

| Institution | Operational Focus | Main Adjustments and Observations |
|---|---|---|
| CBOs | Recruitment and Consent | Staff availability for eligibility verification; differences in reception protocols across sites. |
| | Data Input | Access to devices; differentiated user roles for data entry and service delivery. |
| PHCs | Intake and Eligibility Verification | Clarification of eligibility pathways for undocumented migrants. |
| | Appointment Scheduling | Variation in SH service availability; need to assign responsible personnel. |
| Both | Communication and Coordination | Need to formalize referral pathways and feedback loops between CBOs and PHCs. |

Source: Prepared by the authors, COSMIC project.

Following this validation round, an asynchronous consultation process was conducted via email to finalize site-specific adaptations. This phase enabled each institution to confirm the designated professionals responsible for CBSS implementation and to define their respective operational protocols. As a result, individualized implementation circuits were developed and compiled into an Institutional Implementation Manual, which included visual flowcharts, platform usage guidelines, and communication tools for participant interaction.

Finally, tailored CBSS implementation circuits were finalized for each participating CBO and PHC. For reporting purposes, Figures 3 and 4 present a standardized models that reflect the core operational stages defined during this process. Tailored CBSS implementation circuits were finalized for each participating CBO and PHC. For reporting purposes, Figures 3 and 4 present standardized models that reflect the core operational stages defined during this process.

CBO circuits (Figure 3) include four sequential stages—recruitment, indicators monitoring, counseling, and referral—with adaptations reflecting each organization's specific workflows, staff availability, and user interaction models. For example, CBO 3 conducted participant intake exclusively on a first-come, first-served basis, while CBOs 1 and 2 allowed both walk-ins and scheduled appointments. Differences were also observed in how informed consent was obtained and how sexual health counseling was integrated, with CBO 2 offering it only in specific clinical cases.

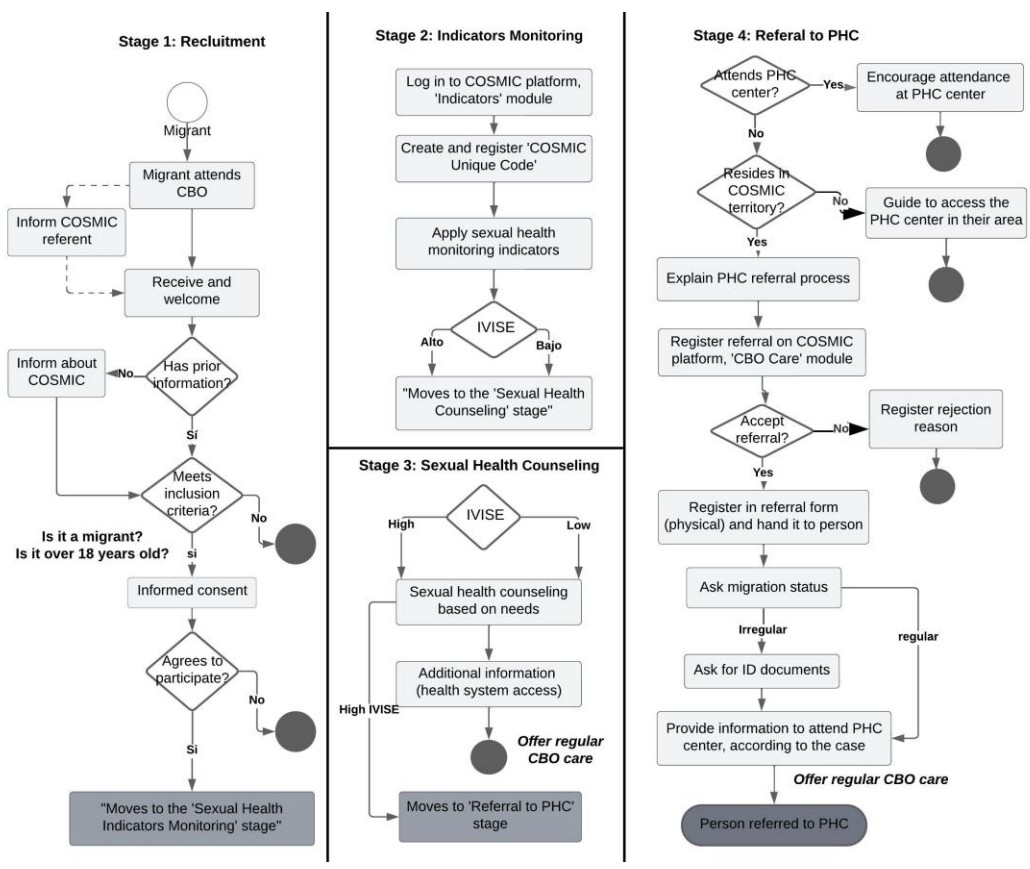

**Figure 3.** Four-stage implementation circuit of the CBOs' CBSS.

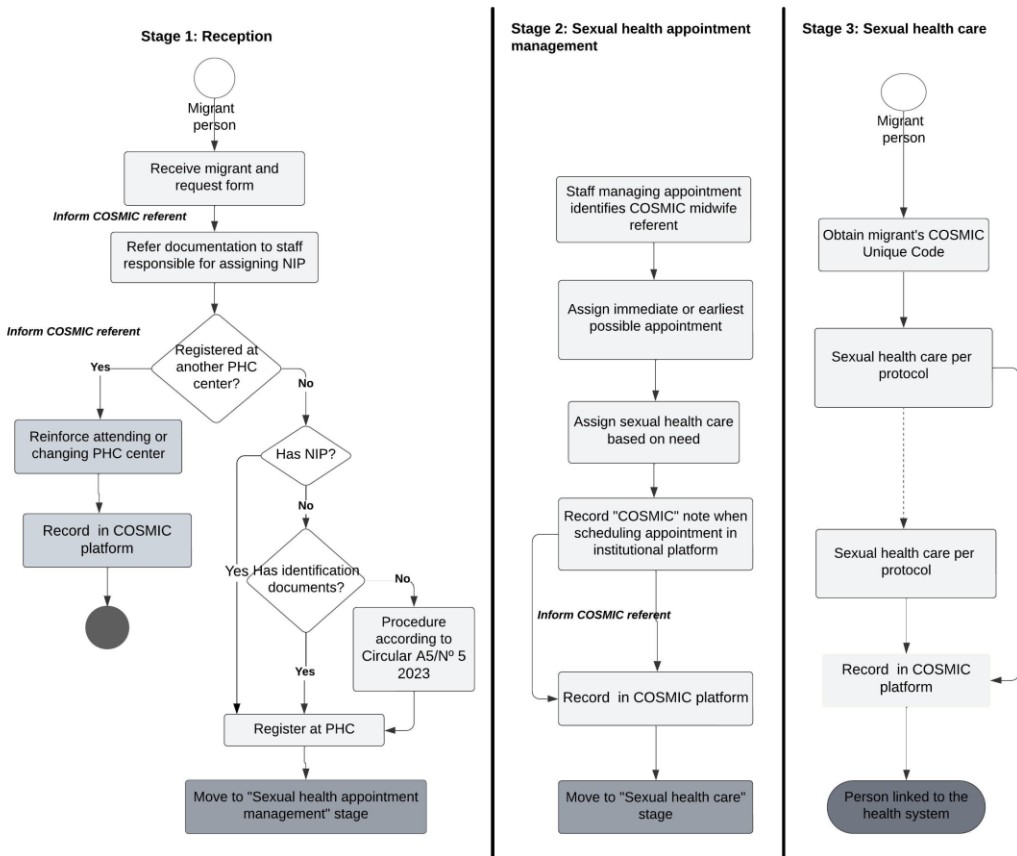

**Figure 4.** Three-stage implementation circuit of the PHCs' CBSS.

PHC circuits (Figure 4), in contrast, follow a more uniform three-stage structure—reception, appointment management, and healthcare delivery. Despite institutional variations, the core flow remained consistent, requiring only adjustments in staff assignments and enrollment procedures based on territorial health jurisdiction and available services.

In conclusion, the second phase of this pre-implementation evaluation focused on translating the findings from the formative stage into an operational design through an iterative and participatory process of developing and refining the CBSS implementation circuits. This phase allowed for the adaptation of the system to the actual capacities, constraints, and workflows of the participating institutions—both CBOs and PHCs—thus strengthening the technical and contextual relevance of the circuits. Key contributions included the identification of operational challenges—such as staff availability, verification of eligibility criteria, referral management, and inter-institutional coordination—as well as the development of tailored solutions for each setting. The resulting circuits offer a context-sensitive implementation model that fosters institutional ownership, improves integration with existing practices, and lays the groundwork for future CBSS deployment, addressing core dimensions of implementation research such as adaptability, acceptability, and sustainability.

## 4. Discussion

This study helped to conduct a pre-implementation evaluation of a CBSS designed to monitor social epidemiological aspects related to SH among the migrant population of two cities from different regions of Chile. This process enabled the development of tailored internal implementation circuits for each participating organization and the establishment of referral strategies to PHCs.

In implementation research (IR), evaluation processes are essential for enhancing the effectiveness of health interventions in real settings, as they help identify and address barriers that hinder proper implementation [21]. This is particularly important given that IR does not control or isolate the conditions or context in which it operates, but rather works within the complex and dynamic environment where interventions take place [28]. A pre-implementation evaluation is crucial because it allows for the early identification of contextual barriers, helps prevent resource or participant losses, and supports timely adaptations to interventions to preserve their effectiveness [21].

From the perspective of IR [22], this paper examines key components of the pre-implementation evaluation framework [21]. It offers a detailed description and characterization of its context and stakeholders, as well as the intervention implementation process and the strategies employed [20,29]. Furthermore, it discusses the adaptations required during implementation [17,28], providing practical insights and evidence that may support researchers and practitioners in tailoring and deploying similar interventions across diverse contexts.

Regarding the identification and addressing of barriers, several studies have identified obstacles to access to SH care among the migrant population [30,31]. The pre-implementation evaluation of this CBSS took into account the complexity of monitoring SH and facilitating access for migrants, given the intersection of violence, and thus sought to reduce some forms of violence and aimed to mitigate some forms of violence through the involvement of CBOs and PHC workers. This included addressing factors such as migratory status, distrust, lack of coverage, and the complexity of the system itself [30]. Although one of its objectives was to strengthen the connection between the public health system and the most vulnerable migrants, this remains a significant challenge, as Chile's public health system is fragmented and hierarchical, hindering access and coordination [32]. Moreover, its segmentation by risk and income generates inequity and inefficiency, concentrating the highest-risk and lowest-income population within the public system [33].

However, IR is expected to be carried out under real-world management and financing conditions [20]. This work is the result of a funded project, and the pre-implementation evaluation made it possible to adapt certain technological and operational conditions to ensure the proper functioning of the CBSS. The literature indicates that such adaptations are common in low- and middle-income countries [34], and given that this is a CBSS, its implementation entails logistical, financial, and technical costs [18].

In the process leading up to the implementation of the CBSS described in this paper, the role of CBOs as valuable sources of public health information should be acknowledged. Accordingly, efforts were made to strengthen their capacities to carry out the surveillance process. Furthermore, measures were implemented to improve the flow of information between CBOs and PHCs, in line with several international guidelines [35].

Moreover, based on the experience gained from this project, several key aspects for ensuring the effectiveness of CBSS should be underscored, including the identification of contextual barriers and the definition of appropriate strategies [36]. One such strategy is integrating the activities related to CBSS operations into the existing responsibilities of community workers. Accordingly, it is essential to streamline and optimize processes to avoid duplicating tasks and facilitate the collection of relevant information [35]. Additionally, involving workers —leveraging their knowledge and experience—in adapting the implementation circuits of the surveillance system has been shown to significantly enhance acceptability and commitment to the CBSS, as supported by the literature in this field [37].

The main lessons and findings of our study lie in the participatory and collaborative nature of the process leading up to the implementation of the CBSS, including the development of the monitoring indicators [23], which involved different CBSS stakeholders. Multidisciplinary and stakeholder participation in the design and implementation of the intervention is a central component of IR [28], which is confirmed in this work. The CBSS was developed through an iterative approach, which required the constant capacity to adapt, both methodologically and operationally, to respond to contingencies that arose locally, in the CBOs or in the public health system.

The work carried out in the development and definition of the CBSS implementation circuits, together with the preparation of manuals and process systematization, strengthens the feasibility and sustainability of CBSS. It also helps document and share the development of an intervention implementation phase in a real environment, without artificially modified conditions, which provides relevant evidence on its applicability and effectiveness in specific contexts [21]. This experience not only contributes to the continuous improvement of the model but also generates transferable learning, facilitating future adaptations to other populations or territories of interest, in line with the dynamic and adaptive IR approaches.

Our study has some limitations. The selection of CBOs and PHCs was based on a formative study; however, for PHCs, centers that were willing to participate were prioritized, which required adapting the CBSS referral circuit to the administrative definitions of the public health system and its areas of responsibility. The reliance on volunteers from CBOs evinced the need for strategies to ensure their sustainability. Nevertheless, the iterative process, consultation with various stakeholders, and active community participation were key strategies for managing our positionality and potential biases. These approaches incorporated diverse perspectives, collectively validated the findings, and allowed the CBSS circuits to be adapted to local realities, thereby enhancing their relevance and legitimacy.

## 5. Conclusions

This pre-implementation evaluation enabled the characterization of institutional capacities, the identification of contextual barriers, and the participatory design of operational circuits for a CBSS focused on SH among the migrant population in Chile. Through a

two-phase qualitative methodology, the study generated empirical evidence that informed the development of differentiated implementation circuits for CBOs and PHCs. These circuits were tailored to institutional capabilities and service flows, enhancing contextual relevance and technical alignment.

The active involvement of local stakeholders throughout this iterative process was crucial to adapting workflows, strengthening institutional ownership, and enhancing acceptability. This approach offers a replicable model for adapting surveillance interventions to complex institutional environments and underscores the value of early-stage evaluations of implementation research. It also contributes to transferable learning for researchers, the community, and health workers engaged in the design of health programs targeting structurally vulnerable populations in comparable contexts.

**Supplementary Materials:** The following supporting information can be downloaded at: https://www.mdpi.com/article/10.3390/sexes6030047/s1, Section S1: Interview Guides for Formative Research; Section S2: Focus Group Interview Guide—Pre-implementation Evaluation of the COSMIC Project; Section S3: Findings of the Pre-implementation Evaluation.

**Author Contributions:** V.S.Á., C.A.P., C.L.D., K.L.A., P.C.H., M.C.-P., C.B.I. and J.B.D. contributed to the design and conception of the study. K.L.A., C.L.D., C.A.P. and V.S.Á. collected the data and carried out the data analysis with support from D.S., D.G. and E.C. C.A.P. wrote the first draft of the manuscript, aided by V.S.Á., C.L.D. and K.L.A. All authors contributed significantly to the interpretation of the data and made critical revisions to the manuscript. All authors have read and agreed to the published version of the manuscript.

**Funding:** This research was conducted as part of the COSMIC project, "Community-based surveillance of social epidemiological aspects linked to sexual health and related communicable diseases among the migrant population in Chile," funded by the National Agency for Research and Development of Chile and the National Fund for Scientific and Technological Development (Fondecyt Regular 1220371).

**Institutional Review Board Statement:** The study was conducted in accordance with the Declaration of Helsinki. The study was approved by the Bioethics Committee, the Vice-Rectorate for Research and Doctoral Programs of the Andrés Bello University (Act No. 017/2022; date of approval: 10 May 2022), the Scientific Ethics Committees of the North Metropolitan Health Services(No. 022/2023, date of approval: 19 July 2023), and the Scientific Ethics Committees of the Antofagasta Health Services(No. 038-23-224/2024; date of approval: 16 January 2024).

**Informed Consent Statement:** Informed consent was obtained from all subjects involved in the study.

**Data Availability Statement:** The data presented in this study are available upon request.

**Acknowledgments:** This work has been carried out within the framework of the PhD programme in Biomedical Research Methodology and Public Health at the Universitat Autònoma de Barcelona. The authors would like to thank all those who contributed to the development of this monitoring system, especially community workers and migrants who generously shared their life stories.

**Conflicts of Interest:** The authors declare no conflicts of interest.

## Abbreviations

The following abbreviations are used in this manuscript:

| | |
|---|---|
| STI | Sexually Transmitted Infection |
| COSMIC | Community-Based Surveillance of Socio-Epidemiological Aspects Linked to Sexual Health and Related Communicable Diseases in Migrant Population in Chile |
| PHCs | Primary Healthcare Center |
| CBOs | Community-Based Organization |
| SH | Sexual Health |
| CBSS | Community-Based Surveillance System |

NIP     Temporary identification number
IVISE   Social–Epidemiological Vulnerability Index
IR      Implementation Research

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
