# Peer review of "Pre-Implementation Evaluation of a Community-Based Surveillance System for Migrants’ Sexual Health in Chile"

_sexes, doi:10.3390/sexes6030047_

Round 1
Reviewer 1 Report
Comments and Suggestions for Authors
I think there is a lot of information presented in this paper and you need to consider focusing on key aspects of the paper and findings, and write in more concise way. Please see attachment for detailed comments and suggestions.

Consider using more plain English and reducing project/study specific jargon
Author Response
Response to Reviewer 1 Comments
|
||
1. Summary |
|
|
Thank you very much for taking the time to review our manuscript. Your comments were highly relevant, and we sincerely appreciate them as they helped to clarify and improve our work. Below, we provide detailed responses along with the corresponding revisions and corrections, which are highlighted in the re-submitted files.
|
||
2. Questions for General Evaluation |
Reviewer’s Evaluation |
Response and Revisions |
Does the introduction provide sufficient background and include all relevant references? |
Can be improved |
We have carefully reviewed and expanded the introduction to ensure it provides sufficient background and includes all relevant references. |
Are all the cited references relevant to the research? |
Yes |
|
Is the research design appropriate? |
Can be improved |
We have made adjustments to improve and clarify the research design in the revised manuscript. |
Are the methods adequately described? |
Can be improved |
We have revised and expanded the Methods section to provide a more detailed and precise description. |
Are the results clearly presented? |
Must be improved |
We have improved the writing and clarity of the Results section in the revised manuscript. |
Are the conclusions supported by the results? |
Yes |
|
3. Point-by-point response to Comments and Suggestions for Authors
|
||
Title: Comments 1: [Please change migrant’s to migrants’] |
||
Response 1: [We agree with this observation and have accordingly corrected the title. |
||
Abstract: Comments 1: [In general. The authors should remember that abstract must be able to stand alone and share the key aspect of the study (background and objectives, methods, results and conclusions and recommendations) in a complete way.] |
||
Response 1: [We appreciate this helpful comment. In response, we have revised the abstract to ensure that it includes all key elements—background and objectives, methods, results, and conclusions—following the structure of a structured abstract, but formatted as a single paragraph without headings, as indicated in the journal’s guidelines for manuscript preparation. We believe that the updated abstract now clearly presents the essential aspects of the study in a concise and coherent manner, aligned with both the reviewer’s suggestion and the journal’s requirements. |
||
Comments 2: [What does ‘implementation circuits’ mean? In general, authors should try to avoid jargon and use simple language and terms that are easy to understand for a broad readership..] |
||
Response 2: We thank the reviewer for this valuable comment. In the context of our study, the term “implementation circuits” specifically refers to flowcharts—a classical tool used to systematically depict the sequence of activities, actors, and decision points within a process. This approach is widely recommended and applied in sanitary and epidemiological surveillance, where it helps illustrate the step-by-step flow of case detection, reporting, investigation, and feedback. Additionally, flowcharts have proven useful in community-based interventions to clarify roles, responsibilities, and ensure a shared understanding among local teams and health managers. In the revised manuscript, we have clarified this concept to avoid ambiguity and make it accessible to a broad readership. Page 5 lines 185 to 192 [“The CBSS “Implementation Circuits” are the operational flows that will guide the participation of individuals within the SVBC, encompassing both the community (OBCs) and PHC levels. From the perspective of public health and epidemiological interventions, the development of these flowcharts constitutes a key strategy to outline processes, roles, and responsibilities, fostering a shared understanding among stakeholders and enabling their adaptation to the specific capacities and contextual characteristics of each institution[23].This phase was carried out through group interviews with key informants from CBOs and PHCs, followed by a group meeting for analysis, feedback, and consensus on the implementation circuits.”] |
||
Comments 3: [Line 31: What are the ‘social epidemiological aspects related to sexual health’ of interest in this study? Can authors be more specific..] |
||
Response 3: [We appreciate and fully agree with this observation. In response, we have incorporated a more specific explanation of the “social epidemiological aspects related to sexual health” in the Introduction section. This clarification is also detailed in our responses to the comments on the main paper]. |
||
Comments 4: [Lines 33-36: It is not clear why and how the 3 CBOs and 2 PHCs were selected? This is because something is missing in the background for the readers to follow, and this does not tie in with objective of the study. What are the roles of CBOs and PHCs relevant to this study?] |
||
Response 4: [We deeply appreciate this comment, which we consider highly pertinent. Many of the points raised regarding the abstract were addressed in the main manuscript. The selection criteria for the CBOs and PHCs are described in the Results section: the CBOs were chosen because they had their own infrastructure, worked specifically in sexual health, and demonstrated the technical capacity to provide counseling in this area—elements essential for the operation of the CBS system. The PHCs, meanwhile, were selected due to their willingness to participate in the COSMIC project and to make the necessary adjustments to their usual processes to accommodate migrant care within the context of the CBS system. Regarding their role in the study, these organizations and centers are fundamental, as the CBS will be implemented directly in these settings, given their close ties to the migrant population.] |
||
Comments 5: [What were the target participants? What methods were used for data collection?] |
||
Response 5: [We appreciate this comment. Information regarding the target participants and the methods used for data collection is detailed in the Materials and Methods section of the manuscript. The target participants were community-based organizations (CBOs) and primary healthcare centers (PHCs) located in Antofagasta and Santiago, Chile] |
||
Comments 6: [Lines 36-37: What does ‘adequate infrastructure and experience’ mean? How was this measured?] |
||
Response 6: We thank the reviewer for this pertinent comment, which we fully acknowledge. In the Results section, we have made a modification to clarify that “adequate infrastructure” refers specifically to having their own premises (headquarters) with a private room or space to receive migrant participants of the SVBC and to conduct sexual health counseling. We believe this addition enhances the precision and understanding of our findings. |
||
Comments 7: [Line 38: Where did ‘stakeholder feedback’ come from since there was no prior mention of data collection from stakeholders?] |
||
Response 7: [We appreciate this important comment. In our study, the term “stakeholders” specifically refers to staff members from the CBOs and PHCs who participated in the process. We have clarified this point in the manuscript to ensure greater precision and to avoid any ambiguity.] |
||
Comments 8: [Was the objective, ‘to describe the context and define the implementation circuits’ answered? This is the question authors should ask themselves at the end of the abstract. I would expect. something like, ‘SVBC is/will be conducted in…., by….. among….to…..(describe the context) and through……(define implementation, the how part)] |
||
Response 8: [We appreciate this valuable comment. In response, we have thoroughly reviewed the abstract and the entire manuscript and made the corresponding adjustments. We consider that the objective has been addressed, as our reference to “describing the context” specifically pertains to the context of the participating institutions.] |
||
Comments 9: [It is also not clear whether the objective was to define implementation circuits or to develop them? Please make this clear.] |
||
Response 9: [We appreciate this important comment. In response, we clarify that the objective of the study was both to define the implementation circuits of the community-based surveillance system and, in doing so, to design, develop, and adapt them. This is because our work did not merely involve conceptually outlining these circuits, but also constructing them through a participatory process with the institutions involved.] |
||
Comments 10: [Lines 41-43: These conclusions are overstated because they are not directly supported by the results presented, e.g., the role of key actors, and effectiveness and sustainability were not assessed based on information provided.] |
||
Response 10: [We appreciate this valuable comment. This point was carefully considered, and as a result, modifications were made both in the abstract and in the conclusion of the manuscript to adjust the statements and align them more precisely with the results presented. The revised abstract reads as follows: “This pre-implementation evaluation helped identify and generate the necessary adaptations for SVBC application. Moreover, our findings underscore how the active involvement of local stakeholders is key to developing implementation circuits that are contextually relevant, acceptable, and appropriately tailored”.] |
||
Main paper Comments 1: [Most comments on the abstract above apply here as well. Please take note.] |
||
Response 1: [Most of the comments on the abstract were carefully addressed, which required making the corresponding adjustments to the manuscript.]” |
||
Comments 2: [Lines 107-109: While it is clear that this was a qualitative study, authors need to describe the specific design of each phase of their study. What exactly did they do? The description should be] |
||
Response 2: We appreciate this valuable comment and fully agree with the observation. In response, we have expanded the Materials and Methods section to provide a more detailed description of the specific design of each phase of the study, as well as the Results section to more precisely reflect what was done in each stage.] |
||
Comments 3: [Line 111: What were the key stakeholders and their roles?.] |
||
Response 3: [We appreciate this important comment. In our study, the term “stakeholders” specifically refers to staff members from the CBOs and PHCs who participated in the process. We have clarified this point in the manuscript to ensure greater precision and to avoid any ambiguity] |
||
Comments 4: [Line 115: First sentence does not belong here but to introduction section.] |
||
Response 4: We agree that the text in question would be more appropriately placed in the Introduction. However, we also consider it important to include the justification for our methodological approach within the Methods section, as explicitly stating the reasons for this choice highlights its relevance and suitability to the study objectives. Additionally, other reviewers requested that we further elaborate on and justify the choice of methods in the manuscript. To address all these valuable comments and enhance the quality of our work, we have incorporated the corresponding modifications into the text.]” |
Comments 5: [Lines 116-118: Here, authors have introduced something new, secondary data analysis. Which data? How was it analyzed? What did the data show? How were the results used?.] |
Response 5: We appreciate this comment, which we consider highly pertinent. Accordingly, we have revised the manuscript to clarify that we did not conduct a secondary data analysis, but rather an exploratory review of the literature (including reports, regulatory frameworks, and legal documents), as well as a search of websites and social media. This process constituted the first phase of the study and aimed to gather background information to understand and contextualize the general setting of community-based organizations in Chile, as well as to preliminarily identify the CBOs and PHCs relevant for the development of the study. Change in 2.3. Phase 1: Formative research “It was carried out through a review of the literature and websites, as well as the qualitative collection of primary information” Change in2.3.1. Identification of CBOs and PHCs “First, CBOs related to migration and SH in the cities of interest were identified. This process included a literature review (including reports, regulatory frameworks, and legal documents) and a search on websites and social networks” |
Comments 6: [Line 129: This statement is not clear and too general. What exactly was done? |
Response 6: We agree with this comment and have therefore revised the entire paragraph to enhance its clarity. The updated version is provided below. “Subsequently, brief structured interviews were conducted with representatives of the initially identified CBOs. During these interviews, the overall objective of the study was presented to participants, and a set of screening questions was applied to determine whether the organizations addressed health issues—particularly sexual health—and whether they worked directly with migrant populations or included migrants among their beneficiaries]” |
Comments 7: [Line 131: What was the source of the search? What was the selection criteria? How many met the criteria? How were the 2 eventually picked?.]
|
Response 7: Thank you for pointing this out. We have changed the paragraph. It should be noted that although this process included organizations identified through both the initial and complementary searches, only the total number of participating CBOs was recorded, without distinguishing between these two groups. This approach aimed to build a broad and relevant pool of organizations aligned with the objectives of the study.
|
Comments 8: [Line 139: ‘….and so on’ does not add value in scientific writing] |
Response 8: Thank you for pointing this out. We agree and, accordingly, we have modified the writing [“among other relevant characteristics”] |
Comments 9: [Lines 192-200: Were these findings from the study or this is information already available? This is probably more suitable for the introduction section.] |
Response 1: Thank you for pointing this out. These are results from the study. The text explains the methodology applied to develop the implementation circuits of the CBSS, whose outcomes are presented in the final section of the Phase 2 results.] |
Comments 2: [ It is unclear to me what social epidemiological aspects are and how they relate to this study.] |
Response 2: We appreciate this valuable comment. In response, we have included a clarification in the introduction of the manuscript (page 1, between lines 105 and 112, “As a foundational step, a set of socio-epidemiological indicators of sexual health was previously developed, understood as social, cultural, economic, and contextual factors that, in interaction with epidemiological patterns, determine the sexual health status of migrant populations”) to explicitly define what we mean by “social epidemiological aspects.” to explicitly define what we mean by “social epidemiological aspects.” Additionally, we have incorporated the corresponding reference so that interested readers may further explore this concept. We believe that these additions strengthen the conceptual foundation of our work and provide clearer context for the reader. |
Comments 3: [Lines 434-435: Again, the conclusion that participation of key stakeholders in the development and definition of the SVBC or interventions with a community approach is valuable in promoting effectiveness and sustainability is overstated as these were not assessed.] |
Response 3: We appreciate this important comment and fully agree with your observation, which is why we made the corresponding adjustments to the conclusions of our work. In response, we carefully revised the conclusion to avoid overstating the role of stakeholder participation in promoting effectiveness and sustainability, as these aspects were not directly assessed in our results, although they are discussed in light of the related scientific literature. The updated conclusion now emphasizes how the involvement of local stakeholders contributed to the development of implementation circuits that were relevant, acceptable, and adapted to the context, without attributing an impact on effectiveness or sustainability. We believe this adjustment directly addresses your observation and aligns the conclusion more precisely with the scope and findings of our study.
|
Comments 4: [Lines 436-439: This conclusion is also generalized and is not directly supported by results.] |
Response 4: We appreciate this valuable comment. In response, we have revised the conclusion to avoid generalized statements that are not directly supported by our results, carefully adjusting it to reflect the specific contributions and scope of our study. Nevertheless, given the nature of this work, we believe it is important to highlight the value of systematically exploring implementation processes and adapting interventions to complex contexts, while ensuring that we do not extrapolate beyond the evidence presented.] |
Comments 4: [Authors should make very specific conclusions and recommendations based on their findings and the objectives of the study] |
Response 4: We appreciate this valuable comment. In response, we have revised the conclusion to include more specific statements and recommendations, directly aligned with the findings and objectives of our study. The updated conclusion can be found on page X, between lines 541 and 553, as follows: “[This study involved a pre-implementation evaluation of a SVBC for monitoring the social epidemiological aspects of SH in migrant population. This process enabled the identification and characterization of four CBOs and two PHCs with potential to participate in its implementation, as well as the recognition of key considerations regarding capacities, barriers, and facilitators for its execution. The active involvement of local stakeholders throughout this iterative process was key to developing and refining the implementation circuits through a participatory approach, enabling the design of operational flows that were contextually appropriate and acceptable, tailored to the specific needs and characteristics of each institution, and thus laying the groundwork for their future deployment. Studies like this underscore of systematically exploring implementation processes and adapting interventions to complex contexts, serving as a valuable reference for researchers and professionals engaged in designing healthcare interventions in in similar settings]” |
4. Response to Comments on the Quality of English Language |
Point 1: “Consider using more plain English and reducing project/study specific jargon” |
Response 1: We appreciate this observation. In response, we have undertaken a thorough review of the manuscript to address instances of ambiguous wording and to correct grammatical inaccuracies. Additionally, the revised manuscript was carefully reviewed by an English language professional to ensure greater linguistic precision and clarity throughout the text. |
5. Additional clarifications |
The comments provided in your review, along with those from the other reviewers, have been extremely valuable for this manuscript. They allowed us to thoroughly re-examine all sections, incorporate elements that strengthen key aspects, and make structural adjustments that enhance the clarity and coherence of the text. Additionally, the manuscript now includes further background information that more precisely contextualizes the origin and significance of this work, which constitutes one stage of a broader research project. It is important to emphasize that, beyond the specific objectives of this study, our intention was to contribute to the field by offering a detailed account of the process involved in conducting a pre-implementation evaluation of a community-based surveillance system, within the framework of implementation research. We believe that documenting these methodological steps—along with the challenges encountered and lessons learned—provides particularly valuable insights for researchers and practitioners who need to plan, adapt, and implement health interventions in complex settings, offering practical guidance that can be extrapolated or serve as a reference for similar initiatives. |

Reviewer 2 Report
Comments and Suggestions for Authors
- Check section 3.1.2. This fits more in the methodology than results. Here, we are interested in the findings, not description of PHCs and CBOs.
- The result section is missing quotations from the interviews. Being a qualitative study, readers may also benefit from the study presentation style. The current style is not the standard practice in qualitative research and should be improved to reflect the actual comments by participants.
- The authors need to show how positionality and possible biases were addressed.
- Authors should indicate how many CBOs were identified from the initial literature/social network search before selecting those providing SH services to migrants.
- There is no information indicating the number of CBO representatives, PHC staff, and other key stakeholders in migration and health were interviewed in total. how many from each CBO/PHC. What categories of stakeholders were included. Why and how were they selected.?
Author Response
Response to Reviewer 2 Comments
|
||
1. Summary |
|
|
Thank you very much for taking the time to review our manuscript. Your comments were highly relevant, and we sincerely appreciate them as they helped to clarify and improve our work. Below, we provide detailed responses along with the corresponding revisions and corrections, which are highlighted in the re-submitted files.
|
||
2. Questions for General Evaluation |
Reviewer’s Evaluation |
Response and Revisions |
Does the introduction provide sufficient background and include all relevant references? |
Yes |
|
Are all the cited references relevant to the research? |
Yes |
|
Is the research design appropriate? |
Can be improved |
We have made adjustments to improve and clarify the research design in the revised manuscript. |
Are the methods adequately described? |
Can be improved |
We have revised and expanded the Methods section to provide a more detailed and precise description. |
Are the results clearly presented? |
Yes |
|
Are the conclusions supported by the results? |
Yes |
|
3. Point-by-point response to Comments and Suggestions for Authors |
||
Comments 1: [Check section 3.1.2. This fits more in the methodology than results. Here, we are interested in the findings, not description of PHCs and CBOs.] |
||
Response 1: [We appreciate this comment and fully agree with the observation. However, we would like to emphasize that one of the objectives of our study was precisely to describe the context, which included, as a result, the selection and characterization of the participating CBOs and PHCs. Nevertheless, we have taken your suggestion into careful consideration and will make the necessary adjustments to improve this section and ensure that it places greater emphasis on the findings] |
||
Comments 2: [The result section is missing quotations from the interviews. Being a qualitative study, readers may also benefit from the study presentation style. The current style is not the standard practice in qualitative research and should be improved to reflect the actual comments by participants] |
||
Response 2: We appreciate this valuable comment. We agree on the importance of including quotations to reflect participants’ voices; however, since the interviews were conducted with the aim of identifying and characterizing institutions, incorporating direct quotes would introduce information that is not further analyzed and would exceed the scope and objectives of this study. |
||
Comments 3: [The authors need to show how positionality and possible biases were addressed.] |
||
Response 3: We greatly appreciate this comment, which we consider highly relevant. In our study, the iterative process, consultation with various stakeholders, and active community participation at all stages of the methodological process were fundamental strategies to address both our own positionality and potential biases. This approach allowed us to incorporate diverse perspectives, collectively validate the findings, and continuously adapt the SVBC implementation circuits to local realities, thereby helping to minimize bias and strengthen the relevance and legitimacy of the process. We have incorporated this point into the Discussion section, where we report the study’s limitations, on page 15 line 534 to 539 [“Nevertheless, the iterative process, consultation with various stakeholders, and active community participation were key strategies for managing our positionality and potential biases. These approaches incorporated diverse perspectives, collectively validated the findings, and allowed the SVBC circuits to be adapted to local realities, thereby enhancing their relevance and legitimacy”.]
|
||
Comments 4: [Authors should indicate how many CBOs were identified from the initial literature/social network search before selecting those providing SH services to migrants]. |
||
Response 4: We greatly appreciate this comment, which we consider highly pertinent. In the manuscript, we have clarified that the literature review constituted an initial exploratory phase, whose primary aim was to conceptualize and characterize CBOs in Chile. This review specifically included reports, regulatory frameworks, and legal documents, as is now detailed in the text. It is important to note that the preliminary identification of institutions was achieved [” Through websites and social networks, 35 CBOs related to migration and health issues were identified: 22 in Santiago and 13 in Antofagasta. Of these, eight in Santiago and five in Antofagasta participated in the characterization process, during which their institutional work and interest in engaging in community research were explored in depth”.] The selection of the eight CBOs in Santiago and five in Antofagasta that participated in the characterization process was primarily based on their willingness and interest in participating, as well as the ability of the research team to successfully establish contact with them. |
||
Comments 5: [There is no information indicating the number of CBO representatives, PHC staff, and other key stakeholders in migration and health were interviewed in total. how many from each CBO/PHC. What categories of stakeholders were included. Why and how were they selected.?] |
||
Response 5: [We are very grateful for this comment and fully agree with it. We have incorporated this information into the manuscript on page 7, between lines 289 and 292: “To establish the linkage between each institution and the COSMIC project team, a representative was designated for each CBO and PHC, most of whom were professionals from the health or social sciences fields. These representatives played a key role in coordinating with the project and within their respective institutions.”] |
||
4. Response to Comments on the Quality of English Language |
||
Point 1: “The English could be improved to more clearly express the research” |
||
Response 1: We appreciate this observation. In response, we have undertaken a thorough review of the manuscript to address instances of ambiguous wording and to correct grammatical inaccuracies. Additionally, the revised manuscript was carefully reviewed by an English language professional to ensure greater linguistic precision and clarity throughout the text. |
||
5. Additional clarifications |
||
The comments provided in your review, along with those from the other reviewers, have been extremely valuable for this manuscript. They allowed us to thoroughly re-examine all sections, incorporate elements that strengthen key aspects, and make structural adjustments that enhance the clarity and coherence of the text. Additionally, the manuscript now includes further background information that more precisely contextualizes the origin and significance of this work, which constitutes one stage of a broader research project. It is important to emphasize that, beyond the specific objectives of this study, our intention was to contribute to the field by offering a detailed account of the process involved in conducting a pre-implementation evaluation of a community-based surveillance system, within the framework of implementation research. We believe that documenting these methodological steps—along with the challenges encountered and lessons learned—provides particularly valuable insights for researchers and practitioners who need to plan, adapt, and implement health interventions in complex settings, offering practical guidance that can be extrapolated or serve as a reference for similar initiatives.
|

Reviewer 3 Report
Comments and Suggestions for Authors
Dear authors,
the article deals with an interesting and relevant topic and results should definitely be published. However, there are some points that need to be revised. Overall, it is difficult to follow the article and there is no common thread. The different text sessions are not sufficiently linked to each other, which is partly due to the fact that relevant content is missing.
The article lacks a comprehensive contextualization and description of the framework.
- A description of the irregular migrant population: Although it is mentioned that the migrant population is high at these two locations, there is no explanation of the characteristics of this migrant population (e.g. age, gender, countries of origin), and the diversity of this population group as well as their living conditions (high significance with regard to vulnerability in relation to the research question) should be explained in more detail in order to emphasize the importance of this study.
- The particularities of the legal framework in Chile are briefly described, but its significance for the healthcare of irregular migrants is not (e.g. access barriers, although other access barriers, e.g. due to the taboo nature of such illnesses, should also be mentioned). This information would also underline the importance of the study.
Methodology:
- However, the description of the method, which is of central importance in this article, is neither fully conclusive (e.g. argumentation as to why this method is best suited, justification of the structure and demonstration of the dependencies of different process steps, descriptions of challenges, etc.) nor sufficiently described.
Results:
- The results also lack various explanations to ensure that the Findings, Discussion and Conclusions are ultimately conclusive. For example, the results on the interest / motivation of the institutions to participate (experiences, identification of problems in practice, etc.).
- Outcomes of the review and feedback? Which adaptations were based on which kind of feedback? How was the framework of these sessions set up to ensure a participatory approach and optimize the results?
- The outcomes of the review and feedback rounds are also not described sufficiently, which means that there are no links to the rest of the text. For example: Which adaptations were based on which kind of feedback? How was the framework of these sessions set up to ensure a participatory approach and optimize the results?
Conclusions
I agree regarding the relevance of this topic, but neither the methodological approach nor the outcomes and challenges were sufficiently described and discussed to achieve the objectives named in the conclusions.
However, the visualizations of the processes and outcomes provide interesting insights and are detailed. However, there is a lack of in-depth insights into the basis on which these models were created.
I recommend a comprehensive review of this article and am convinced that the topic is important and that interesting results are available.

The article contains ambiguous wording and incorrect English grammar in some passages, which should be corrected.
Author Response
Response to Reviewer 3 Comments
|
||
1. Summary |
|
|
Thank you very much for taking the time to review our manuscript. Your comments were highly relevant, and we sincerely appreciate them as they helped to clarify and improve our work. Below, we provide detailed responses along with the corresponding revisions and corrections, which are highlighted in the re-submitted files.
|
||
2. Questions for General Evaluation |
Reviewer’s Evaluation |
Response and Revisions |
Does the introduction provide sufficient background and include all relevant references? |
Must be improved |
We have carefully reviewed and expanded the introduction to ensure it provides sufficient background and includes all relevant references. |
Are all the cited references relevant to the research? |
Can be improved |
We have reviewed and strengthened the references cited in the manuscript |
Is the research design appropriate? |
Must be improved |
We have made adjustments to improve and clarify the research design in the revised manuscript. |
Are the methods adequately described? |
Can be improved |
We have revised and expanded the Methods section to provide a more detailed and precise description. |
Are the results clearly presented? |
Must be improved |
We have improved the writing and clarity of the Results section in the revised manuscript. |
Are the conclusions supported by the results? |
Can be improved |
We have ensured that the conclusions are clearly supported by the results in the revised manuscript. |
3. Point-by-point response to Comments and Suggestions for Authors |
||
Comments 1: [Overall, it is difficult to follow the article and there is no common thread. The different text sessions are not sufficiently linked to each other, which is partly due to the fact that relevant content is missing.] |
||
Response 1: [We sincerely appreciate your observation regarding the lack of a clear thread and the insufficient connection between the different sections of the manuscript. We have thoroughly revised the entire article and implemented various modifications aimed at enhancing the coherence and integration of its contents. In particular, we have incorporated and strengthened elements that more effectively link the different sections, thereby ensuring a more consistent and well-structured argumentative flow throughout the manuscript.] |
||
Comments 2: [The article lacks a comprehensive contextualization and description of the framework] |
||
Response 2: [Type your response here and mark your revisions in red] Thank you for pointing this out. I/We agree with this comment. Therefore, I/we have….[Explain what change you have made. Mention exactly where in the revised manuscript this change can be found – page number, paragraph, and line.] |
||
Comments 3: [A description of the irregular migrant population: Although it is mentioned that the migrant population is high at these two locations, there is no explanation of the characteristics of this migrant population (e.g. age, gender, countries of origin), and the diversity of this population group as well as their living conditions (high significance with regard to vulnerability in relation to the research question) should be explained in more detail in order to emphasize the importance of this study.] |
||
Response 3: We appreciate your valuable observation regarding the need to provide a more detailed description of the characteristics of the irregular migrant population and the legal framework governing their access to healthcare. In response to this comment, we have incorporated additional background information that allows for a more precise characterization of the migrant population. These elements are detailed on page 2, lines 63 to 66, “Additionally, there has been a marked increase in the proportion of migrants with irregular migration status, which grew from 0.8% in 2018 to 17.6% in 2023 [5]. More than 70% of the migrant population originates from other Latin American countries, primarily Venezuela (38%) and Peru (13.6%)”. |
||
Comments 4: [The particularities of the legal framework in Chile are briefly described, but its significance for the healthcare of irregular migrants is not (e.g. access barriers, although other access barriers, e.g. due to the taboo nature of such illnesses, should also be mentioned). This information would also underline the importance of the study. |
||
Response 4: We thank you for this important comment. In the first version of the manuscript, we chose to synthesize certain contextual aspects to provide more space and emphasis on the description of the results. It is worth noting that in Chile there are very few studies that address the specific barriers to healthcare access for irregular migrants from a regional or national perspective, which is precisely one of the aspects we seek to explore through the surveillance system implemented in the COSMIC project. Nevertheless, we fully agree on the need to provide additional background to better contextualize the study. Therefore, we have incorporated more detailed information on the relevance of the Chilean legal framework and other barriers to healthcare access (including cultural and social aspects) on page 2, lines 72 to 75, in the paragraph beginning with “[However, several barriers to accessing and utilizing healthcare services still exist for this population [7], as evidenced by 16.4% of migrants lacking health insurance (compared to 1.7% of the local population) and an equal proportion not receiving care for recent health issues, versus 9.5% among Chilean-born individuals]”. |
||
Comments 5: [Methodology: However, the description of the method, which is of central importance in this article, is neither fully conclusive (e.g. argumentation as to why this method is best suited, justification of the structure and demonstration of the dependencies of different process steps, descriptions of challenges, etc.) nor sufficiently described.] |
||
Response 5: [We appreciate your observation regarding the methodology, which we consider a central aspect of this study. In response, we have thoroughly reviewed the manuscript, expanding the methodological description and clarifying the rationale behind our decision to combine a formative research approach with a flexible process for constructing implementation pathways, in line with the principles of implementation research. This has been incorporated into the Materials and Methods section, under Study Design, on page 3, lines 117 to 120 (“The methodological approach is grounded in the principles of implementation research, which aims to generate context-specific evidence to inform the design, adaptation, and scaling of interventions in real-world settings.”) ”), and is also addressed in the description of the two study phases. This strategy allows for contextualizing the design, adaptation, and scaling of the community-based surveillance system, identifying key stakeholders, mapping resources, recognizing barriers, and facilitating processes tailored to the characteristics of the participating organizations and primary care centers. Additionally, we have improved the structure of the results section to enhance clarity and better illustrate the connection between the methodological components and the findings obtained. Furthermore, we included a section at the end of the introduction that elaborates on the origin of this manuscript within the framework of a broader research project, located on page 1, lines X to X (“This work is part of the COSMIC project (FONDECYT, Regulation No. 1220371), which aims to develop a system to monitor social epidemiological and cultural aspects a related to SH and communicable diseases in the migrant population in Chile , as well as to fa-cilitate their linkage with primary healthcare centers (PHCs), within the framework of implementation research [19]. As a foundational step, a set of socio-epidemiological in-dicators of sexual health was previously developed, understood as social, cultural, eco-nomic, and contextual factors that, in interaction with epidemiological patterns, determine the sexual health status of migrant populations. Building on this groundwork, a Com-munity-Based Surveillance System (CBSS) was designed to monitor these indicators in real time. This article evaluates the pre-implementation phase of the CBSS, with the aim of describing its context and defining its implementation circuits, a key component within implementation research”). |
||
Comments 6: [Results: The results also lack various explanations to ensure that the Findings, Discussion and Conclusions are ultimately conclusive. For example, the results on the interest / motivation of the institutions to participate (experiences, identification of problems in practice, etc.)] |
||
Response 6: We sincerely appreciate this comment and fully agree with your observation. We recognize that, in our effort to provide a detailed account of the process, we did not delve deeply enough into certain findings. Consequently, we have incorporated changes in the Results section to expand on these aspects and made adjustments to the Conclusion to more accurately reflect the contributions of the study. These changes are underlined in the updated manuscript we have submitted. |
||
Comments 7: [Results:, Outcomes of the review and feedback? Which adaptations were based on which kind of feedback? How was the framework of these sessions set up to ensure a participatory approach and optimize the results?] |
||
Response 7: We appreciate this valuable comment, which raises important questions related to the results presented in section “3.2.2.2 Group review and feedback.” The findings derived from the review and feedback process are detailed in section “3.2.2.3 Final circuits.” Your observation allowed us to recognize the need not only to reorganize the presentation of the results in the manuscript but also to further elaborate on them. While our initial intention was to distinguish the different stages of the pre-implementation process of the CBS in order to contribute from our experience, we understand that the structure employed may have hindered a clear understanding. Therefore, we have adjusted the organization of the text and expanded the results to enhance clarity and facilitate reading]” |
||
Comments 8: [Conclusions: I agree regarding the relevance of this topic, but neither the methodological approach nor the outcomes and challenges were sufficiently described and discussed to achieve the objectives named in the conclusions. However, the visualizations of the processes and outcomes provide interesting insights and are detailed. However, there is a lack of in-depth insights into the basis on which these models were created. I recommend a comprehensive review of this article and am convinced that the topic is important and that interesting results are available.] |
||
Response 8: [We sincerely appreciate this comment, which we consider extremely valuable for the authors. Both this observation and the overall review have allowed us to identify weaknesses in the writing of the manuscript, as our initial effort to synthesize the results led to insufficient emphasis on key aspects of the findings in the first version. In response, we conducted a thorough and comprehensive revision of the manuscript, adding information to the introduction, the materials and methods section, and especially the results, with the aim of strengthening the presentation and overall understanding of the study]” |
||
4. Response to Comments on the Quality of English Language |
||
Point 1: “The article contains ambiguous wording and incorrect English grammar in some passages, which should be corrected” |
||
Response 1: We appreciate this observation. In response, we have undertaken a thorough review of the manuscript to address instances of ambiguous wording and to correct grammatical inaccuracies. Additionally, the revised manuscript was carefully reviewed by an English language professional to ensure greater linguistic precision and clarity throughout the text. |
||
5. Additional clarifications |
||
The comments provided in your review, along with those from the other reviewers, have been extremely valuable for this manuscript. They allowed us to thoroughly re-examine all sections, incorporate elements that strengthen key aspects, and make structural adjustments that enhance the clarity and coherence of the text. Additionally, the manuscript now includes further background information that more precisely contextualizes the origin and significance of this work, which constitutes one stage of a broader research project. It is important to emphasize that, beyond the specific objectives of this study, our intention was to contribute to the field by offering a detailed account of the process involved in conducting a pre-implementation evaluation of a community-based surveillance system, within the framework of implementation research. We believe that documenting these methodological steps—along with the challenges encountered and lessons learned—provides particularly valuable insights for researchers and practitioners who need to plan, adapt, and implement health interventions in complex settings, offering practical guidance that can be extrapolated or serve as a reference for similar initiatives. |
